# Obesity Measured via Body Mass Index May Be Associated with Increased Incidence but Not Worse Outcomes of Immune-Mediated Diarrhea and Colitis

**DOI:** 10.3390/cancers15082329

**Published:** 2023-04-17

**Authors:** Miho Kono, Malek Shatila, Guofan Xu, Yang Lu, Antony Mathew, Wasay Mohajir, Krishnavathana Varatharajalu, Wei Qiao, Anusha S. Thomas, Yinghong Wang

**Affiliations:** 1Department of Gastroenterology, Hepatology and Nutrition, The University of Texas MD Anderson Cancer Center, Houston, TX 77030, USA; 2Department of Nuclear Medicine, The University of Texas MD Anderson Cancer Center, Houston, TX 77030, USA; 3Department of Internal Medicine, The University of Texas Health Science Center at Houston, Houston, TX 77030, USA; 4Department of Biostatistics, The University of Texas MD Anderson Cancer Center, Houston, TX 77030, USA

**Keywords:** obesity, BMI, immune checkpoint inhibitor, colitis

## Abstract

**Simple Summary:**

The association of obesity with a host of comorbidities and its role in cancer have been studied extensively, but its influence on cancer treatments is not well understood. In particular, little is known about the interplay between obesity and immune checkpoint inhibitor (ICI)-induced immune-related adverse events (irAEs). This retrospective single-center study explored this relationship in 202 cancer patients with ICI exposure who developed gastrointestinal irAEs and had existing data on their body mass index (BMI) and visceral fat as measured by CT. Lower BMI was interestingly found to correlate with a more severe disease course. Aside from that, obesity was not found to significantly alter the course of ICI-mediated diarrhea and colitis, nor did it impact the overall survival of this population. Importantly, this study also supports the use of BMI as an indicator of adiposity in cancer patients, as higher BMI values were strongly associated with increased visceral fat on CT imaging.

**Abstract:**

Obesity defined by high body mass index (BMI) has traditionally been associated with gastrointestinal inflammatory processes but has recently been correlated with better survival in patients receiving immune checkpoint inhibitors (ICI). We sought to investigate the association between BMI and immune-mediated diarrhea and colitis (IMDC) outcomes and whether BMI reflects body fat content on abdominal imaging. This retrospective, single-center study included cancer patients with ICI exposure who developed IMDC and had BMI and abdominal computed tomography (CT) obtained within 30 days before initiating ICI from April 2011 to December 2019. BMI was categorized as <25, ≥25 but <30, and ≥30. Visceral fat area (VFA), subcutaneous fat area (SFA), total fat area (TFA: VFA+SFA), and visceral to subcutaneous fat (V/S) ratio were obtained from CT at the umbilical level. Our sample comprised 202 patients; 127 patients (62.9%) received CTLA-4 monotherapy or a combination, and 75 (37.1%) received PD-1/PD-L1 monotherapy. Higher BMIs ≥ 30 were associated with a higher incidence of IMDC than BMIs ≤ 25 (11.4% vs. 7.9%, respectively; *p* = 0.029). Higher grades of colitis (grade 3–4) correlated with lower BMI (*p* = 0.03). BMI level was not associated with other IMDC characteristics or did not influence overall survival (*p* = 0.83). BMI is strongly correlated with VFA, SFA, and TFA (*p* < 0.0001). Higher BMI at ICI initiation was linked to a higher incidence of IMDC but did not appear to affect outcomes. BMI strongly correlated with body fat parameters measured by abdominal imaging, suggesting its reliability as an obesity index.

## 1. Introduction

The obesity crisis is a longstanding global health issue. In 2015, over a third of the world’s population was found to be overweight. Overweight individuals suffer from a host of sequelae [1]. They are at increased risk of developing conditions such as diabetes mellitus, cardiovascular disease, musculoskeletal disorders, and poor neurological health [2,3,4,5]. Obesity is regarded as a leading cause of preventable deaths globally, causing a predicted 400,000 deaths annually [6]. Obesity has also been found to play a considerable role in the incidence, prognosis, and survival of a less preventable leading cause of death: cancer [7]. To date, there has been much research on the role of obesity in cancer pathophysiology. It has been associated with the development of cancer in at least 13 different anatomical locations and has a variable influence on cancer survival depending on the cancer type [8,9,10,11,12]. However, given the speed at which cancer treatment is evolving in the modern era, there is an unfortunate dearth of research involving the effects of obesity on the efficacy of cancer therapeutics.

Immune checkpoint inhibitors (ICIs)—namely, cytotoxic T-lymphocyte antigen 4 (CTLA-4) and programmed death 1/programmed death ligand 1 (PD-1/PD-L1) inhibitors—have revolutionized the treatment of advanced cancers by improving patient survival across a wide range of tumors [13]. However, ICIs can give rise to an autoimmune response in the various organ systems of the body through the selective upregulation of tumor antigen-specific T-cells, resulting in immune-related adverse events (irAEs) [14]. irAEs in the gastrointestinal (GI) system, referred to as immune-mediated diarrhea and colitis (IMDC), are typically present as diarrhea that may necessitate hospitalization and further treatment. IMDC is among the most common irAE and a frequent reason for ICI discontinuation [15,16]. Given the success of ICIs and the treatment disruption by IMDC, it is important to identify the risk factors contributing to the development and severity of this disease entity; obesity, in particular, is a highly relevant factor in this regard.

Body mass index (BMI) evaluates an individual’s weight relative to their height and is a widely employed index of obesity [17]. Recent evidence suggests that BMI may serve as an emerging prognostic factor among cancer patients with ICI exposure [18,19,20,21]. A multicenter study of cancer patients treated with anti-PD-1/PD-L1 showed that overweight and obese patients (BMI ≥ 25) experienced significantly more irAEs, particularly GI-related ones, compared to non-overweight patients (BMI < 25) (55.6% vs. 25.2%) [20,21]. This study also found that overweight and obese patients had significantly longer overall survival (OS) and progression-free survival (PFS) compared to non-overweight patients [20]. Similarly, another pooled analysis of 4090 cancer patients treated with ICIs suggested that patients with a BMI of ≥25 had a significantly higher risk of developing irAEs compared to patients with a BMI of <25 [21]. Finally, a meta-analysis of 3768 cancer patients with ICI exposure showed that higher BMI was associated with improved PFS and OS while exhibiting similar mortality and tumor progression rates compared to normal BMI, regardless of tumor type [22].

Despite the high specificity of the standard cut-off value in defining obesity, BMI has a low sensitivity in identifying adiposity and is, therefore, insufficient to delineate visceral fat composition and discriminate between types of adiposities [23,24]. There are established quantitative measures of intra-abdominal fat volume, such as visceral fat area (VFA), subcutaneous fat area (SFA), and total fat area (TFA: sum of VFA and SFA), which can be obtained from computed tomography (CT) imaging and used to evaluate obesity and obesity-related risks [25,26,27]. Emerging evidence also supports the utility of body fat measurement by computed imaging in studying the efficacy of ICI therapy in cancer patients [28,29,30,31]. Notably, higher SFA, VFA/SFA (V/S) ratio, and TFA among advanced cancer patients treated with ICIs were associated with increased OS and PFS [28,30,32]. Meanwhile, VFA was found not to have an effect on OS and PFS in a sample of non-small cell lung cancer patients on ICIs [31].

The current literature on the impact of obesity on IMDC outcomes in cancer patients is still lacking. In this study, we aim to investigate the association between obesity as measured by BMI with IMDC incidence, disease course, and outcome and to assess whether BMI is a valid reflection of adiposity in this population using CT-generated fat measures.

## 2. Materials and Methods

### 2.1. Patient and Clinical Data Selection

We retrospectively analyzed patients at a tertiary cancer center who were treated with ICI from April 2011 to December 2019 and met the following inclusion criteria: (1) aged >18 years old, (2) had an established cancer diagnosis and received ICIs (PD-L1 inhibitors, PD-1 inhibitors, or CTLA-4 inhibitors) as monotherapy or as combination therapy, (3) had a diagnosis of IMDC, and (4) underwent abdominal CT imaging within 14 days before initiating ICI therapy. This study was approved by the institutional review board with a waiver of patient’s informed consent.

All patient data were collected through institutional electronic medical databases. Baseline demographics included age, sex, race/ethnicity, comorbidities, and the Charlson comorbidity index [33]. Oncologic information such as cancer type, cancer stage (using the 8th edition American Joint Committee on Cancer staging manual), ICI therapy type (CTLA-4 monotherapy, PD-1/PD-L1 monotherapy, or combination of both), and date of ICI initiation was also gathered, alongside mortality and OS. We broadly categorized malignancies as melanoma, genitourinary cancer, lung cancer, and others (GI cancer, head and neck, endocrine, and hematologic malignancy). Cancer staging for hematological malignancies was not included.

### 2.2. IMDC Characteristics and Outcomes Assessed

The collected characteristics of IMDC were date of onset, peak diarrhea, and colitis grades at the time of diagnosis based on the National Cancer Institute’s Common Terminology Criteria for Adverse Events (CTCAE; version 5.0), duration of IMDC symptoms, use of intravenous steroids, duration of steroid therapy, and use of non-steroidal immunosuppressants, i.e., infliximab and vedolizumab. The collected outcomes of IMDC included hospitalization, duration of hospitalization, recurrence rate, and clinical remission rate. Clinical remission of IMDC symptoms was defined as a sustained resolution of the symptoms to grade 1 or lower. Clinical response of IMDC was defined as an improvement of symptoms to a lower grade than the initial presentation but not meeting the criteria of clinical remission.

### 2.3. Obesity Measurement

BMI was calculated from patient height and body weight (BMI = weight in kilograms divided by the square of height in meters) within 30 days before ICI initiation. BMI values were divided into 3 groups: underweight/normal weight (BMI < 25 kg/m^2^), overweight (BMI ≥ 25 kg/m^2^ but <30 kg/m^2^), and obesity (BMI ≥ 30). VFA, SFA, TFA, and the V/S ratio were obtained from CT scans at the level of the umbilicus as an intra-body fat index (Figure 1) based on established standards [34]. MIM Software was used for the measurement of intra-body fat area from radiologic imaging. Subcutaneous fat was defined as the extraperitoneal fat between skin and muscle. Visceral fat was defined as intraperitoneal fat with the same density on imaging as the subcutaneous fat layer. CT scans were analyzed using pre-established thresholds for fat of VFA and SFA (−150 to −50 Hounsfield units). All measurements were carried out by two experienced radiologists.

### 2.4. Statistical Analysis

SPSS version 24.0 software was used to perform the statistical analyses. Mann–Whitney U and Kruskal–Wallis tests were used to compare continuous variables between groups. Chi-square and Fisher exact tests were used to evaluate the association between categorical variables. Univariate logistical regression was used to estimate the difference in IMDC incidence among BMI subgroups. The Kaplan–Meier method and log-rank test were used to measure OS. OS was defined as the time from the initiation of ICI therapy to the date of last follow-up or death. Pearson correlation analysis was applied to assess the correlation between BMI and visceral body fat parameters. *p* values of <0.05 were considered statistically significant.

## 3. Results

### 3.1. Patient Baseline Characteristics

A total of 573 IMDC patients had abdominal and pelvic CT scans at the cancer center from 2011 to 2020. Among them, 202 patients were included in this study (patient selection flow chart in Figure 2). Our cohort had a median age of 61 years (interquartile range [IQR] 49–70 years). A majority (89.6%) were white, and a majority were male (67.3%). The most common cancer type was melanoma (44.1%), followed by genitourinary cancer (29.7%) and lung cancer (7.9%). Most patients had a diagnosis of stage IV cancer (87.1%). Regarding therapy, 127 (62.9%) received CTLA-4–based therapy (alone or with PD-1/PD-L1 therapy), and 75 (37.1%) received PD-1/PD-L1 monotherapy. The median follow-up time was 28.8 months (IQR 14.2–48.8 months), and all-cause mortality was 44.1%. The median BMI was 28.1 (IQR 24.4–32.6). Further information regarding patient characteristics can be found in Table 1.

### 3.2. Characteristics of IMDC Stratified by BMI

The incidence of IMDC was analyzed and compared among different BMI groups in the base number of 2139 patients who had BMI data available and received ICI, with abdominal CT scan done within 14 days prior to ICI initiation, matching the IMDC cohort. A BMI ≥ 30 was found to have the highest incidence of IMDC at 11.4%, compared to patients with a BMI between 25–30 (9.1%; *p* = 0.158) and ≤25 (7.9%; *p* = 0.029) (Appendix A). Characteristics of IMDC were summarized and compared between the three BMI groups (Table 2). Individuals with lower BMI were found to have higher CTCAE severity of colitis (*p* = 0.03). The other IMDC characteristics, including time of IMDC onset, duration of symptoms, and use of steroids and non-steroidal immunosuppressants, were similar between BMI groups.

### 3.3. Outcomes of IMDC Stratified by BMI

The outcomes of IMDC by BMI group are summarized in Table 2. Overall, approximately 60% of patients required hospital admission, with a median length of stay of 5–7 days depending on BMI. Clinical remission of IMDC on medical treatment was achieved in 70% of patients, and the colitis recurrence rate was 15%–25% between groups within the follow-up time of 25–33 months from IMDC diagnosis. There was no significant difference observed in the comparison of these outcome variables between the three BMI groups. On OS analysis, BMI level did not appear to affect the survival duration (*p* = 0.829) (Figure 3).

### 3.4. Correlation between BMI and Visceral Body Fat Parameters

Correlation analysis demonstrated strong associations between BMI level and the levels of SFA (*p* < 0.001), VFA (*p* < 0.0001), and TFA (*p* < 0.0001) (Table 3).

## 4. Discussion

Despite the tremendous improvements in survival in a wide range of metastatic solid tumors treated with ICIs, ICI exposure introduces the risk of irAEs, which often require ICI discontinuation and hospitalization [16,35]. An understanding of the potential clinical and demographic risk factors associated with irAEs and their severity is critically needed to improve clinical outcomes. To our knowledge, our study has, for the first time, demonstrated that higher BMIs may be associated with an increased risk for IMDC, with lower rather than higher BMI presenting higher colitis grades. There was no discernable impact of BMI on IMDC outcomes or OS.

Previous studies have investigated the association between BMI and body fat index in different patient populations [36,37,38,39]. BMI compares an individual’s height relative to their weight as a convenient proxy measure for assessing body fat and has been used in pediatric and adult populations to screen for obesity [17,40,41,42]. BMI is a predictive factor for the risk of developing comorbidities such as diabetes mellitus and cardiovascular disease [43,44]. However, BMI has recently come under scrutiny for not being an accurate assessment of obesity [45,46], and whether BMI characterizes body adiposity remains controversial [17,36,47,48]. Body adiposity, particularly visceral adipose tissue, is suggested to be the main culprit in obesity-related complications and is, therefore, an important parameter [49]. In cancer patients specifically, visceral fat and the corresponding insulin resistance and chronic low-grade inflammation play an important role in the disease course, more so than BMI [50,51,52]. Still, most studies on cancer patients continue to use BMI as a measure of obesity despite its contentious association with visceral fat [53,54,55,56,57]. In this study, we compared BMI measures with body fat indices obtained through the gold standard of CT imaging and found strong correlations (*p* < 0.0001), which supports our use of BMI for the accompanying analyses. However, further validation of BMI as a substitute for CT-generated body fat indices is needed.

High BMI presents an “obesity paradox” for survival among cancer patients whereby it increases the risk of esophageal, colon, and renal cancers while also being a favorable prognostic factor in several solid cancers, such as non–small cell lung cancer, colorectal cancer, and gastric cancer [9,10,11,12]. Recently, BMI was found to be a promising clinical marker for predicting the development of irAEs and their outcomes [18,19,20,21]. Though this tool has extended the possibilities for nutritional assessment in routine clinical care for cancer patients, the relationship between BMI and IMDC—which frequently leads to ICI discontinuation—has not been clear. Our study among cancer patients who developed IMDC importantly showed a possible association between increased BMI ≥30 and the risk of developing colitis compared to patients with a BMI ≤ 25. This association disappears when looking at patients with a BMI in between, likely due to the large variety inherent in that range. This is in line with results from previous research showing an association between obesity and the development of irAEs [19,20]. Furthermore, we found that colitis severity was inversely correlated with BMI. Higher grades of colitis (grade 3–4) were correlated with lower BMI, and BMI had no significant impact on OS, which contradicts previous reports on the association between higher BMI and worse irAE outcomes, including colitis [20,58]. This discrepancy may result from multiple confounding factors, e.g., our unique cohort of patients with IMDC, or the specific type of cancer studied. However, these studies investigated irAEs in all systems, not GI irAEs in particular, which could suggest unique associations between IMDC and obesity. We suspect the benefit to OS from IMDC may overcome the influence of BMI on survival among this cohort, which may also explain the non-significant association of BMI with OS [59]. Further studies, including a control group without IMDC, may clarify the impact of BMI on patients’ survival.

Cancer patients who develop IMDC usually have diarrhea and colitis; this clinical presentation is highly reminiscent of inflammatory bowel disease [60,61]. Particularly in Crohn’s disease, obesity has been associated with a worse disease course [62,63]. Higher VFA was associated with an increased risk of surgery for Crohn’s disease and with early post-operative Crohn’s recurrence [61,64,65]. Moreover, higher VFA was an independent predictive factor for delayed intestinal mucosal healing among patients with Crohn’s disease that required infliximab [66]. Distinct characteristics of adipose distribution may thus contribute to the pathophysiology of Crohn’s disease and possibly IMDC. Further analysis is still needed to evaluate the association between the pattern of adipose distribution and inflammatory bowel conditions.

The impact of obesity on the incidence and mortality of a wide range of cancer types has already been recognized for up to 13 different cancers [13]. The current hypothesis is that the increased volume of visceral fat promotes tumorigenesis and increases cancer risk through an adverse impact on inflammation and metabolism [67,68,69]. This is based on the knowledge that inflammatory cells are more prevalent in visceral fat compared to subcutaneous fat, are more metabolically active, insulin resistant, and sensitive to lipolysis, and are, therefore, more pro-inflammatory and pro-tumorigenic [70,71,72,73,74]. One suggested reason for such phenomena is the existence of non-coding RNA (ncRNA includes long ncRNA and microRNA). It has been discovered that a majority of our genome is transcribed into ncRNA that serves various roles in regulating gene expression, including immune checkpoint genes [75]. The level of expression of these ncRNA has been implicated in the development and prognosis of certain tumors and chemo- and immunotherapy resistance [76]. A few studies have hypothesized that ncRNA expression is altered in obesity [77,78], potentially explaining the current link between obesity and cancer. However, there is no conclusive evidence of this yet. Given the wide range of existing ncRNAs, each with a differential effect on tumorigenesis and immunotherapy resistance, it would be interesting to explore this mechanistic link between BMI and immunotherapy efficacy, adverse event rate, and disease course in the future. Additionally, although the predictive value of various cutoffs for adiposity have been studied for certain medical conditions, e.g., metabolic syndrome, there are no standardized criteria for VFA, SFA, or TFA regarding cancer risk, the effect of cancer treatment, and irAEs, which warrants additional research studies [79,80]. Additionally, it is worth looking into the impact of sarcopenia and its relationship with adiposity, as previous studies have shown that sarcopenia may be associated with worse survival among various cancer types [81,82,83].

Gut microbiota has been associated with obesity and immunotherapy outcomes among cancer patients. Obesity can produce chronic pro-inflammatory states, which can cause a dysfunctional gut barrier and lipopolysaccharide leakage [84]. Immunotherapy also induces gut mucosal damage by compromising barrier integrity, which increases the gut invasion of bacteria. Clinically, multiple studies have demonstrated that certain microbiome compositions (e.g., low *Faecalibacterium*, Ruminococcaceae, and other Firmicutes with enriched *Bacteroides*) among cancer patients on ICI are associated with better survival, while other compositions were associated with increased bowel inflammation [85,86,87]. Indeed, the gut microbiome has a significant immunoregulatory effect, and the types of bacteria present may play a crucial role in mediating both bowel inflammatory responses and possibly more distant, systemic inflammatory responses [88,89,90]. Interestingly, previous research has shown that the gut microbiome may also play a role in the development of obesity, and restoration of healthy commensal bacteria via fecal microbiota transplantation has shown promising results for improving outcomes in cancer, refractory IMDC, and obesity [91,92,93,94,95,96,97,98]. Given the complex interplay between obesity, the inflammatory cascade, gut microbiomes, cancer, and cancer treatment, it is especially challenging to delineate a causative relationship and any interactive effects among these variables. Further studies are, therefore, needed to clarify the role of the microbiome and fecal microbiota transplantation in treating obesity in the general population and IMDC in an obese cancer population.

Our study has several limitations. First, this is a retrospective single-center study with all the inherent limitations that stem from this study design. The data availability in patients’ charts could limit the accurate collection of all the details related to IMDC episodes. Second, additional cancer therapies after ICI termination were not included in this study, which could potentially confound the IMDC outcomes and cancer survival. Third, the lack of a control group without IMDC rendered unfeasible the evaluation of the impact of obesity on IMDC incidence and overall cancer outcome. Fourth, we included only patients who had an abdominal CT scan in this study window with images amenable for fat measurement, which could certainly lead to a selection bias. Finally, dietary factors and the microbiome could play a critical role in the parameters investigated in this study; however, we were not able to collect this information retrospectively.

## 5. Conclusions

IMDC has been recognized as one of the most common irAEs in cancer patients on ICI treatment. The association between obesity and IMDC and its disease course has not been studied. We found that patients with BMIs on the higher end (at or above 30) may be at increased risk of developing IMDC, with lower BMIs (at or below 25) associated with a decreased risk but higher colitis severity. BMI had no impact on survival. BMI categorization has a good correlation with body fat index measurement via imaging and could be a useful and convenient tool in oncologic practice to predict various risks associated with cancer and ICI therapy. Future prospective trials are needed to further elucidate the effect of adiposity on the characteristics and outcomes of IMDC in cancer patients on ICIs.

## Figures and Tables

**Figure 1 cancers-15-02329-f001:**
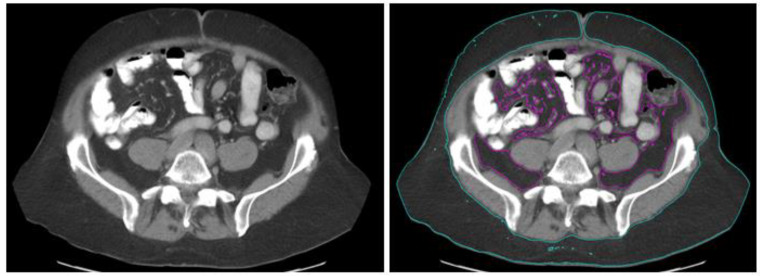
Measurement of intra-abdominal fat area by use of computed tomography. (**Left**): Initial computed tomography images. (**Right**): Intra-abdominal adipose tissue areas. Regions of purple and blue color indicate visceral (89.4 cm^2^) and subcutaneous (246.6 cm^2^) adipose tissue, respectively.

**Figure 2 cancers-15-02329-f002:**
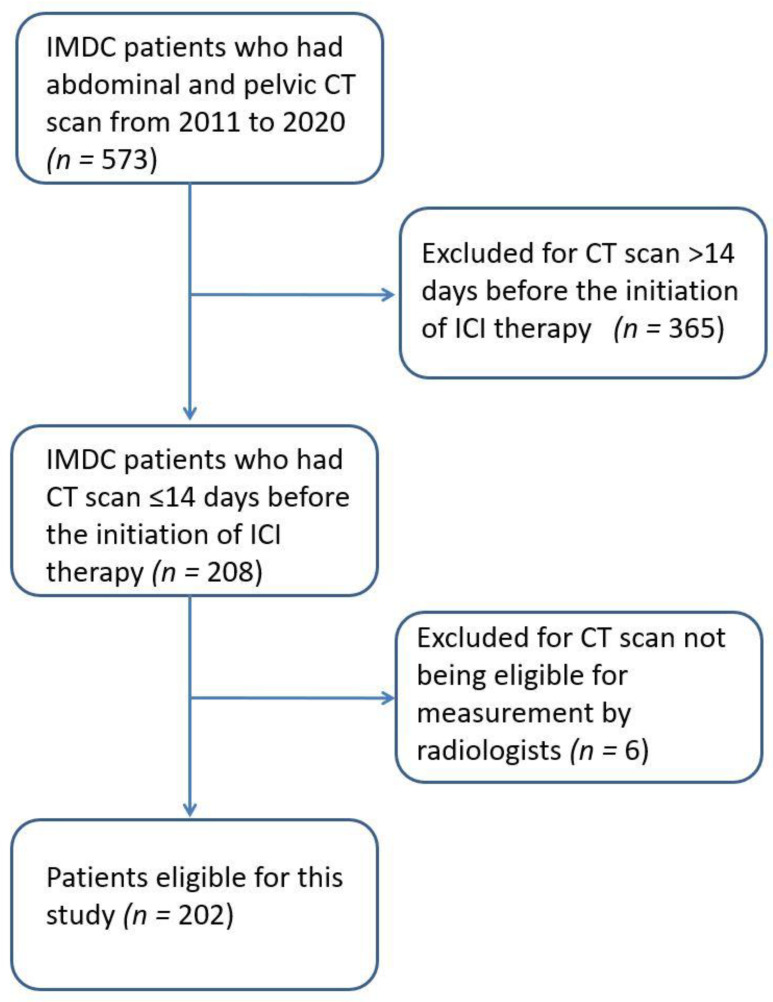
Patient selection flow chart.

**Figure 3 cancers-15-02329-f003:**
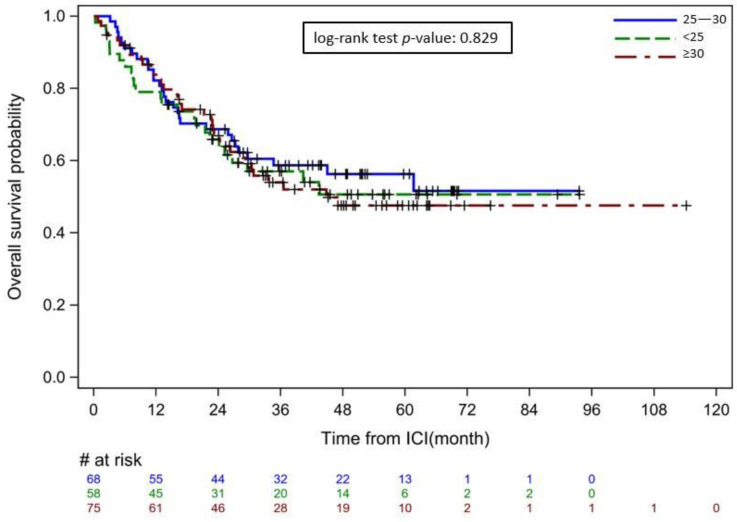
Overall survival of patients with immune-mediated diarrhea and colitis by BMI group Starting point: immune checkpoint inhibitor initiation. Ending point: last follow-up or death from any cause. # refers to the number of patients at risk, + indicates cases censored due to loss to follow-up.

**Table 1 cancers-15-02329-t001:** Demographic characteristics.

Characteristic	Value (*N* = 202)
Age, median (IQR), years	61 (49–70)
Race/ethnicity, *n* (%)	
White	181 (89.6%)
Others	21 (10.4%)
Male sex, *n* (%)	136 (67.3%)
Cancer type, *n* (%)	
Melanoma	89 (44.1%)
Genitourinary cancer	60 (29.7%)
Lung cancer	16 (7.9%)
Others	37 (18.3%)
Cancer stage, *n* (%)	
III	24 (11.9%)
IV	176 (87.1%)
Other	2 (1.0%)
Metabolic syndrome, *n* (%)	
Hypertension	120 (59.4%)
Hyperlipidemia	90 (44.6%)
Diabetes	65 (32.2%)
Charlson comorbidity index, median (IQR)	8 (6–9)
ICI therapy types, *n* (%)	
CTLA-4 monotherapy	57 (28.2%)
PD-1/PD-L1 monotherapy	75 (37.1%)
Combination of CTLA-4 and PD-1/PD-L1	70 (34.7%)
Follow-up duration from ICI initiation to last follow-up/death, median (IQR), months	28.8 (14.2–48.8)
All-cause mortality, *n* (%)	89 (44.1%)

Abbreviations: IQR, interquartile range; ICI, immune checkpoint inhibitor.

**Table 2 cancers-15-02329-t002:** Characteristics and outcomes of IMDC stratified by BMI.

IMDC Measures		BMI		*p* Value
	<25 *N* = 58	≥25 but <30 *N* = 68	≥30 *N* = 76	
Incidence of GI irAEs	7.96%	9.12%	11.41%	0.085 ^†^
Characteristics of IMDC				
Duration between first dose of ICI to IMDC, median (IQR), months	1.9 (0.9–5.3)	1.8 (1.0–3.5)	2.4 (1.4–4.2)	0.33
Peak grade of colitis, *n* (%)				0.03
≤2	34 (59%)	43 (63%)	60 (79%)	
3–4	24 (41%)	25 (37%)	16 (21%)	
Peak grade of diarrhea, *n* (%)				0.38
≤2	22 (38%)	31 (46%)	38 (50%)	
3–4	36 (62%)	37 (54%)	38 (50%)	
Duration of IMDC symptoms, median (IQR), days	19.0 (7.0–41.0)	21.0 (10.0–45.0)	18.0 (6.0–37.0)	0.32
Duration of steroid therapy, median (IQR), days	37.0 (19.8–52.5)	40.0 (17.0–60.0)	30.0 (20.5–70.0)	0.91
Use of intravenous steroids, *n* (%)	25 (43%)	37 (54%)	35 (46%)	0.45
Use of non-steroidal immunosuppressants, *n* (%)	26 (45%)	28 (41%)	26 (34%)	0.48
**Outcomes of IMDC**				
Hospitalization of IMDC, *n* (%)	36 (62%)	41 (60%)	46 (61%)	0.98
Duration of hospitalization, median (IQR), days	7.0 (5.0–12.0)	6.0 (4.0–11.0)	5.0 (3.0–8.0)	0.40
Clinical remission after IMDC treatment, *n* (%)	40 (69%)	47 (69%)	56 (74%)	0.43
Recurrent IMDC, *n* (%)	14 (24%)	14 (21%)	12 (16%)	0.62
Overall survival, median (IQR), months	25.5 (13.4–45.0)	33.1 (15.0–51.7)	29.0 (15.9–47.8)	0.32

Abbreviations: IMDC, immune-mediated diarrhea and colitis; BMI, body mass index; IQR, interquartile range; ICI, immune checkpoint inhibitor. ^†^ See Appendix A. for further details.

**Table 3 cancers-15-02329-t003:** Correlation analysis between BMI and body fat parameters on imaging.

Parameter	Pearson Correlation Coefficient	*p* Value
SFA	0.635	<0.0001
VFA	0.579	<0.0001
TFA	0.673	<0.0001
V/S ratio	0.115	0.105

Abbreviations: SFA, subcutaneous fat area; VFA, visceral fat area; TFA, total fat area; V/S ratio, VFA/SFA ratio.

## Data Availability

All collected data is available upon request.

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
