# Peer review of "Obesity Measured via Body Mass Index May Be Associated with Increased Incidence but Not Worse Outcomes of Immune-Mediated Diarrhea and Colitis"

_cancers, 2023, doi:10.3390/cancers15082329_

Round 1

Reviewer 1 Report

It is a very well-written manuscript covering an emerging topic of BMI association with the irAEs, particularly colitis and diarrhoea.

The only mini comment:

At East coast we commonly observe patients with better response rates (short-term) with higher BMI >32. Have you observed the similar findings in your retrospective trial.

In addition, some studies suggest that more in-depth analysis is needed to determine the exact role of obesity on ICI efficacy and or safety. In particular, non-coding RNA and bystander T cells were proposed to be associated with the higher incidence of irAEs. Could you please comment on it in discussion.

Author Response

It is a very well-written manuscript covering an emerging topic of BMI association with the irAEs, particularly colitis and diarrhoea.

The only mini comment:

At East coast we commonly observe patients with better response rates (short-term) with higher BMI >32. Have you observed the similar findings in your retrospective trial.

Response: Thank you for the insight! Our primary endpoints (overall mortality and clinical remission of colitis at last follow-up) did not really allow us to explore short-term cancer response rates but our data do suggest that there may be a numerical (but not statistically significant) advantage to higher BMI in some regards to colitis disease course. Particularly, our results show that patients with a BMI ≥30 had a lower grade of colitis (79% grade 2 or below vs. 61% of those with BMI <30; p=0.03), required a shorter duration of steroid therapy (30 days vs. 37-40 days in those with BMI <30), and a higher rate of clinical remission of colitis by study end (74% vs. 69%). It would be interesting to explore this in a larger-scale study.

In addition, some studies suggest that more in-depth analysis is needed to determine the exact role of obesity on ICI efficacy and or safety. In particular, non-coding RNA and bystander T cells were proposed to be associated with the higher incidence of irAEs. Could you please comment on it in discussion.

Response: Thank you for bringing our attention to this subject! We reviewed the literature and added some description on this topic and references as reviewer suggested. We noted that some miRNA upregulated PD-1 expression and increased tumor resistance to immunotherapy while other types of non-coding RNA might improve immunotherapy efficacy. We also found a few studies that hypothesize that obesity may alter non-coding RNA expression, and that RNA may also increase the risk of obesity. A brief section has been added in the discussion (found below) to comment on this.

Revised Discussion:

This is based on the knowledge that inflammatory cells are more prevalent in visceral fat compared to subcutaneous fat, are more metabolically active, insulin resistant, and sensitive to lipolysis, and are therefore more pro-inflammatory and pro-tumorigenic [70-74]. One suggested reason for such phenomena is the existence of non-coding RNA (ncRNA; includes long ncRNA and microRNA). It’s been discovered that a majority of human genome is transcribed into ncRNA that serve various roles in regulating gene expression, including immune checkpoint genes (75). The level of expression of these ncRNA has been implicated in the development and prognosis of certain tumors and in chemo- and immunotherapy resistance (76). A few studies have hypothesized that ncRNA expression is altered in obesity (77, 78), potentially explaining the current link between obesity and cancer. However, there is no conclusive evidence of this yet. Given the wide range of existing ncRNAs each with a differential effect on tumorigenesis and immunotherapy resistance, it would be interesting to explore this mechanistic link between BMI and immunotherapy efficacy, adverse event rate, and disease course in the future.

Reviewer 2 Report

Kono et al. present an interesting study describing the association, or lack of, between obesity measures and outcomes of immune-mediated colitis.

They analyzed retrospectively 202 cancer patients with ICI exposure who developed gastrointestinal irAEs and had existing data on their body mass index (BMI) and visceral fat as measured by CT.

The results are clearly presented. The study is interesting because obesity is often associated with comorbidities.

I have only one major comment:

- It is unclear to this reviewer, why only patients with IMDC were included in the study. I understand that the authors conclude that obesity is not a risk factor for worse outcome of IMDC; however, it is unclear from this study, whether obesity is per se a risk factor for developing IMDC. If the authors have access to a database with information on all patients with obesity-related information, I would strongly advise to make one additional analysis on the potential association between obesity and development of IMDC, not severity of IMDC

Author Response

Kono et al. present an interesting study describing the association, or lack of, between obesity measures and outcomes of immune-mediated colitis.

They analyzed retrospectively 202 cancer patients with ICI exposure who developed gastrointestinal irAEs and had existing data on their body mass index (BMI) and visceral fat as measured by CT.

The results are clearly presented. The study is interesting because obesity is often associated with comorbidities.

I have only one major comment:

- It is unclear to this reviewer, why only patients with IMDC were included in the study. I understand that the authors conclude that obesity is not a risk factor for worse outcome of IMDC; however, it is unclear from this study, whether obesity is per se a risk factor for developing IMDC. If the authors have access to a database with information on all patients with obesity-related information, I would strongly advise to make one additional analysis on the potential association between obesity and development of IMDC, not severity of IMDC

Response: Thank you so much for your kind and succinct description of our study. We agree with your valuable input that adding a measure of the incidence of IMDC among different BMI subgroups would greatly improve the quality of our work. Unfortunately, since one of the main outcomes we were exploring was the association between BMI and visceral fat measurement via body imaging, our patient selection criteria was restricted to patients who developed IMDC and had CT imaging done within 14 days of checkpoint inhibitor initiation. Given this selection bias, running the additional analysis would need to change the focus and the inclusion criteria of the study and underestimate the real incidence of IMDC. In addition, the base sample size of over 17000 patients on ICI in the study window from our institution will make the data collection infeasible. We will keep this suggestion in mind for future projects as it would be an interesting link to explore.

Reviewer 3 Report

Comments to the Authors

The manuscript Miho Kono et al. entitled “Obesity Measured via Body Mass Index is Not Associated with Worse Outcomes of Immune-Mediated Diarrhea and Colitis” (Manuscript ID: cancers-2249669) seems interesting. There are points which need to be addressed.

1.      Recently, sarcopenia is also focused as a predictor for the prognosis of cancers. Reviewers are interested in muscle volume of the patients in this study and the association with outcomes. The measurement of muscle volume, assessment of  sarcopenia, and their analyses are thought difficult, but authors can touch on topics of the association among fat volume, sarcopenia, and BMI for the prognosis in cancer patients.

2.      Although reviewers are not sure about the limit number of references in submission guidelines of this journal, the number of references in this manuscript seems too many. For example, is it necessary to cite references published more than 15-20 years ago? Reviewers would recommend authors to examine the necessity of many of them.

Author Response

Reviewer 3 comment:

  1. Recently, sarcopenia is also focused as a predictor for the prognosis of cancers. Reviewers are interested in muscle volume of the patients in this study and the association with outcomes. The measurement of muscle volume, assessment of sarcopenia, and their analyses are thought difficult, but authors can touch on topics of the association among fat volume, sarcopenia, and BMI for the prognosis in cancer patients.

Response: Thank you for your feedback. While the focus of our manuscript was on adiposity, we agree that sarcopenia would be another interesting variable to look at. Unfortunately, we don’t have that data readily available to attempt that analysis, which as you mention may be difficult to conduct. However, we have added a sentence in the discussion to touch on this subject as below:

“Also worth looking into is the impact of sarcopenia and its relationship with adiposity, as previous studies have shown that sarcopenia may be associated with worse survival among various cancer types [81-83].”

  1. Although reviewers are not sure about the limit number of references in submission guidelines of this journal, the number of references in this manuscript seems too many. For example, is it necessary to cite references published more than 15-20 years ago? Reviewers would recommend authors to examine the necessity of many of them.

Response: Thank you for your input. We agree that there is a large number of references included in this manuscript. Regarding the manuscripts published >20 years ago, many of them were used to define specific terms or introduce particular subjects. Since obesity has been a public health issue for many years now, most of the original papers describing the entity originated around 20 years ago which we hope to reiterate to the readers. The authors reviewed journal guidelines and see that there is no set number limit on references.

Round 2

Reviewer 2 Report

The authors answered that "Given this selection bias, running the additional analysis would need to change the focus and the inclusion criteria of the study and underestimate the real incidence of IMDC. In addition, the base sample size of over 17000 patients on ICI in the study window from our institution will make the data collection infeasible"

This reviewer does not understand how performing one additional analysis on patients with CT imaging done within 14 days of checkpoint inhibitor initiation would underestimate the real incidence of IMDC. In fact, the authors state in their answer that "our patient selection criteria was restricted to patients who developed IMDC and had CT imaging done within 14 days of checkpoint inhibitor initiation". It would simply require including all patients who had CT imaging done within 14 days of checkpoint inhibitor initiation, regardless IMDC development; this analysis should be very feasible, given the retrospective nature of this study but provided approval from the institution/IRB (if applicable). Having or not CT imaging done within 14 days of checkpoint inhibitor initiation should not, logically, have any relation with the development of IMDC.

Author Response

Reviewer comment:

The authors answered that "Given this selection bias, running the additional analysis would need to change the focus and the inclusion criteria of the study and underestimate the real incidence of IMDC. In addition, the base sample size of over 17000 patients on ICI in the study window from our institution will make the data collection infeasible" This reviewer does not understand how performing one additional analysis on patients with CT imaging done within 14 days of checkpoint inhibitor initiation would underestimate the real incidence of IMDC. In fact, the authors state in their answer that "our patient selection criteria was restricted to patients who developed IMDC and had CT imaging done within 14 days of checkpoint inhibitor initiation". It would simply require including all patients who had CT imaging done within 14 days of checkpoint inhibitor initiation, regardless IMDC development; this analysis should be very feasible, given the retrospective nature of this study but provided approval from the institution/IRB (if applicable). Having or not CT imaging done within 14 days of checkpoint inhibitor initiation should not, logically, have any relation with the development of IMDC.

Response: First, we would like to thank you for your valuable input. We were able to pull the data requested based on the same criteria and calculate an incidence for IMDC among different BMI subgroups. We found that patients with a BMI ≥ 30 were significantly more likely to develop IMDC than patients with a BMI ≤ 25 (11.4% incidence vs. 9.4%; p=0.029). We really appreciate your advice, and have edited the manuscript abstract, methods, results, conclusions, and tables accordingly.

Round 3

Reviewer 2 Report

The authors have performed one additional analysis that show a distinct risk of developing GI-irAE depending on BMI. I would suggest the authors to make additional changes to the abstract, discussion and perhaps even the title to better reflect the new findings (Higher BMI associated with the risk of developing GI-irAE but not with the severity of GI-irAE, or something similar)

Author Response

Reviewer comment:

The authors have performed one additional analysis that show a distinct risk of developing GI-irAE depending on BMI. I would suggest the authors to make additional changes to the abstract, discussion and perhaps even the title to better reflect the new findings (Higher BMI associated with the risk of developing GI-irAE but not with the severity of GI-irAE, or something similar)

Response: Thank you for your valuable input on this manuscript. We have edited the abstract and title accordingly to highlight the new findings. We have also made some minor additions to the discussion.